# A Conservative Image Boundary Extraction Method with Application to the ILM Tumor Surgery

## Abstract

While infant lymphatic malformation tumors are benign, they are very difficult to remove. The removal process is very delicate and requires the retention of as much healthy tissue as possible. Commonly utilized boundary extraction methods aim to extract boundaries covering the vast majority of the target area which remove more healthy tissue than is desirable. This paper presents a conservative image boundary extraction (CIBE) approach with well-designed iterative boundary shrinkage procedures which are applied to computerized tomography (CT) images for use in ILM tumor resection operations. CIBE incorporates three primary concepts: Fuzzy Degree, Pixel Deepness and Boundary Smoothness. The proposed algorithm first converts the marked CT image into a 0-1 image matrix. Then it shrinks the boundary according to the estimated PD and BS indices for the image in an iterative fashion until the boundary smoothness meets the desired level. Empirical analysis demonstrates that the smooth, conservative tumor boundaries are obtained using the CIBE algorithm. The proposed method can also be easily extended to the three dimensional studies.

## 1 Introduction

The lymphatic malformation is derived from the congenital malformation of the lymphatic system. It is common in children and adolescents. According to de Serres et al. (1995), the Infant Lymphatic Malformation (ILM) is an abnormal growth that usually appears on a baby's neck or head. It consists of one or more tumors and tends to grow larger over time. Although the ILM tumors are not malignant, however those located in the oral and maxillofacial areas can seriously impact children's maxillofacial development with a rapid expanding and may cause bleeding or infection. In the worst cases, they can cause the deterioration in appearance and organ function, such as pain, swelling, and breathing difficulties.

One of the most common treatments is the surgical resection method based on the CT image of the lesion area. In principle, all lesion regions are expected to be removed. However, it is hard to remove the lymphatic malformation tumors altogether. According to Mandel (2004), abnormal lymphatic vessels usually have complex internal structures, large size and irregular shape. Resecting the diseased area completely may cause damages to the wall of lymphatic vessels, which further injures the surrounding normal tissues. Thus, a conservative extracted boundary is desired to retain as much healthy tissues as possible in the ILM tumor excision surgery. Motivated by this, we aim to develop an analytical method to conservatively extract the smooth boundary of the core area of the lymphatic malformation tumor according to the patient's CT image diagram. However, most available boundary extraction methods aim to extract boundaries covering the vast majority of the target area. There has been some previous works that achieved certain conservative levels for the boundary extraction. Abrantes & Marques (1996) described a class of constrained clustering algorithms for the object boundary extraction that includes several well-known algorithms proposed in different fields. Catté et al. (1992) presented the image selective smoothing and the edge detection methods based on nonlinear diffusion. Sun & Takayama (1999) came up the method that the border edge is conservatively smoothed on adaptive quadrilateral grids. However, our scenario requires absolute conservatism that there is no healthy tissue within the extracted core lesion area.

In the ILM tumor boundary extraction study, the lesion area is marked in red manually by the surgeons, thus the pixel values of the CT image are meaningless. Therefore, in our experiment we first convert the raw CT image data into a 0-1 data matrix for further analysis, where the 0-point group is the none-lesion area, and 1-point group is the lesion tissue. There are three challenges in the ILM tumor boundary extraction: (1) extremely disordered boundary. At the contact area between the none-lesion regions and the lesion regions, the 0-points and 1-points are mixed and interleaved with each other. The boundary is extremely irregular and unclear; (2) unusual request for depicting boundary. The ILM tumor requires the extracted boundary must be conservative covering the core part of the lesion area without hurting any healthy tissues in the surgery; (3) high standards for the boundary smoothness. The extracted boundary is expected to be smooth since the boundary is used as navigation guidance in the automatic surgery system.

To overcome these challenges, we propose an conservative image boundary extraction (CIBE) method to obtain a contractive and smooth tumor boundary in an iterative way. The proposed method appropriately retain the core lesion area of the tumor. In our method, we innovatively propose three indices, which are the Fuzzy Degree ($FD$), the Target Pixel Deepness ($PD$) and the Boundary Smoothness ($BS$). After obtaining the 0-1 matrix, we first calculate the $PD$ index of each point in the 1-point group, and then get the $BS$ index for the extracted images. If the $BS$ index of the extracted boundary is less than the desired shrinkage level, it indicates that the boundary is smooth enough. Otherwise, the points whose $PD$ indices are less than the shrinkage parameter are eliminated from the 1-point group and the remaining points are entered into the next extraction cycle. The CIBE algorithm runs iteratively until we get a clear, smooth, conservative image boundary. The flowchart of the proposed iterative algorithm is shown in Figure 1. The experiments show that the proposed CIBE algorithm extracts a smooth boundary around the focal region without any healthy tissues. There are many similar scenarios in applications, such as the cell morphological analysis and the polluted area analysis. Our proposed method can be easily applied to these applications with similar conservative boundary extraction requests.

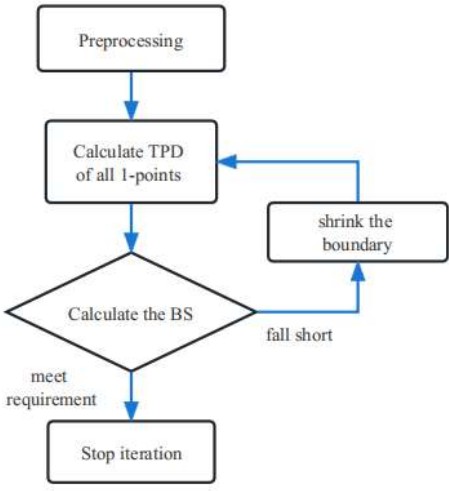

Figure 1: The flowchart of the CIBE algorithm

The rest of the article is organized as follows. Section 2 introduces the notation and model, the proposed iterative algorithm, and the definition of $FD$, $PD$ and $BS$. We analyze the infant lymphatic malformation tumor data in Section 3. Finally, the discussion in Section 4 concludes the article.

## 2  METHOD AND NOTATION

Let $M = \{(x, y, z)|x, y \in \mathbb{Z}^+; z \in [0, 1]\}$ be a three dimensional image, where $x$ and $y$ are the coordinates of the location of the point, and $z$ is the pixel value of the point $(x, y)$. $M_1 = \{(x, y, z)|x, y \in \mathbb{Z}^+; z \neq 0\}$ is the set of all none-zero pixels of $M$. $M_0 = M_1^c$ is the set of all zero pixels of $M$. Let $f : z|x, y \to \{0, 1\}$ be a mapping function that transforms $M$ into a data two-dimensional $0 - 1$ matrix, where

$$f(z|x, y) = \begin{cases} 1, & \text{if } z \neq 0 \\ 0, & \text{if } z = 0 \end{cases}.$$

Next, we introduce the concept of the Fuzzy Degree ($FD$), which describes the exposure level of the point $(x, y) \in M$ in the class $M$. We denote the set of the $k$-nearest neighbors of the point $(x, y)$ as $KNB(x, y)$, where

$$KNB(x, y) = \{(x'_j, y'_j) \,|(x'_j, y'_j) \in M \text{ with the smallest}$$
$$d_{(x,y),(x',y')}, \; j = 1, \ldots, k\},$$

where $d_{(x,y),(x',y')} = \sqrt{(x - x')^2 + (y - y')^2}$ is the Euclidean distance. Let $k$ represent the scope of algorithmic monitoring and the diameter of the neighborhood. When $k$ is a small value, the decision to remove a particular point depends solely on the state of its neighborhood. A larger $k$ expands the range of neighborhood, which will have higher possibilities to include more class-0 points into the neighborhood. The choice of $k$ is not freewheeling. In a two-dimensional pixel image, when $k = 1$, it is not possible to locate a unique nearest neighbor, because the nearest neighboring pixels in the up, down, left, and right positions around the target pixel have the same minimum distance. Therefore, the algorithm adjusts $k$ to 4 to avoid the uncertainties of the nearest neighbors. Based on the same logic, the nearest neighbor measurements are the same for $k \in \{12, 13, \ldots, 19, 20\}$ in the same image, as $k \in \{12, 13, \ldots, 19, 20\}$ will be automatically adjusted to $k = 20$. Therefore, the unique settings of the parameter $k$ are $k \in \{4, 8, 12, 20, 24, 28, 36, 44, 48 \ldots\}$. Within a small range, the variation of $k$ has limited impact on the results. However, when $k$ is large, the entire image will be completely cleared out (the grayscale vales of all pixels will be set to be 0). This is because when the image is shrunk into an area with relatively small diameters, a large search radius will include a lot of class-0 points into the nearest neighborhood, which will cause the underestimate of the PD index of all class-1 points within the search range, and thus intensifies the removal power of the algorithm and all points are removed. $KNB(x, y)$ can be divided into two sets according to the class label of the neighbors, which are the set of class-1 neighbors $KNB_1(x, y) = KNB(x, y) \cap M_1$ and the set of class-0 neighbors $KNB_0(x, y) = KNB(x, y) \cap M_0$.

**Definition 1**  *Fuzzy Degree ($FD$)*

*The fuzzy degree of the point $(x, y)$ is*

$$FD(x, y) = (1 - \lambda)f(x, y) + \lambda \frac{NK_1(x, y)}{k},$$

*where $FD(x, y) \in [0, 1]$, $NK_1(x, y) = \|KNB_1(x, y)\|$ is the number of $k$-nearest Class-1 data points in the neighborhood of $(x, y)$. $\lambda \in [0, 1]$ weakens the classification information that the data carries.*

The fuzzy degree $FD(x, y)$ denotes the degree of the point $(x, y)$ belonging to Class-1. $\lambda$ represents the impact of the nearest neighbors on the fuzziness of the classification information of the target point $(x, y)$. It balances the tradeoff between the classification information carried on the data point $(x, y)$ and its $k$-nearest neighbors. Especially, when $\lambda = 0$, $FD(x, y)$ coincides with the class label of the data; when $\lambda = 1$, $FD(x, y)$ is solely influenced by the classification information of $k$-nearest neighbors.

**Definition 2**  *Pixel Deepness ($PD$)*

*The Pixel Deepness of a target point $(x, y)$ is*

$$PD(x,y) = \frac{\sum_{(x',y')\in KNB(x,y)} FD(x',y')\frac{1}{d^2_{(x,y),(x',y')}}}{\sum_{(x',y')\in KNB(x,y)} \frac{1}{d^2_{(x,y),(x',y')}}}.$$

*where $PD \in [0,1]$, and $FD$ is the fuzzy degree of the point $(x',y')$.*

The $PD$ index measures the deepness of a class-1 point located in the class-1 cluster. A point located at the center of the class-1 cluster will have its $PD$ index close to 1, while a point at the edge of the cluster area will have its $PD$ index close to 0. The statistical significance of $PD$ index is the probability of being inside the boundary. A smaller $PD$ indicates a smaller probability of being inside the boundary. Point with small $PD$ index will be excluded from the class-1 domain. Especially for the class-1 pixels near the boundaries, their $PD$ indexes tend to be small, which makes them susceptible to be removed. In a CT image, from the $PD$ index we can tell which pixels are located deep into the lesion area. In our proposed iterative algorithm, points with small $PD$ indexes will be labeled as non-pathological tissues at each iteration.

**Definition 3** *Boundary Smoothness ($BS$)*

*The Boundary Smoothness of a image is*

$$BS(M_1) = \min_{(x,y)\in M_1} PD(x,y).$$

*where $BS \in [0,1]$, and $M_1$ is a set of all non-zero pixels.*

In our scenario, the smoothness of the extracted boundary has to be guaranteed. The $BS$ index is derived from the most prominent pixel points in the excised area. The larger the $BS$ index is, the smoother the boundary will be. In general, a boundary smoothness close to 0.5 indicates that the boundary is sufficiently smooth. If the $BS$ index is too low, it indicates that the boundary is not smooth enough, and there still exists some redundant pixels in the excised region.

Before introducing our algorithm, we need to define a shrinkage parameter $s \in [0,1]$. The shrinkage parameter $s$ is a hyperparameter represents the shrinkage degree of the boundary. In each iteration of the shrinking process, points with $PD$ index less than or equal to $s$ are removed. When $s = 0$, there is no shrinkage and all points are retained. When $s = 1$, the entire image will be cleared out and the grayscale values of all points are set to 0. Suppose there is a point located on an ideal flat surface, where its neighboring class-0 points and class-1 points are symmetric about the plane, then the $PD$ index of this point is 0.5. In reality, the boundaries are always curved surfaces, and the $PD$ index of the pixels on the boundary are likely to be less than 0.5. Therefore, if $s > 0.5$, the final image will be transformed into a fully zero image. The bigger the $s$ is, the higher the degree of shrinkage of the boundary will be, and the image of the extracted area will be smaller and the boundary will be smoother.

Algorithm 1 is our proposed conservative image boundary extraction (CIBE) algorithm. In our experiments, the converted grayscale data matrix of a CT image with an area marked in red is provided, where the grayscale value in the data matrix represents the strength of the marks. We transform the grayscle value of all none-zero pixels into 1 in the first step, and denote these none-zero pixels as the class-1 pixels. Then we calculate the $FD$ index, the $PD$ index, and $BS$ index of all class-1 pixels to get the smoothness degree of the current extracted boundary. If the BS index of the current extracted boundary is greater than the shrinkage parameter $s$, it means that the boundary is smooth enough. Otherwise, remove the pixels whose $PD$ indexes are less than the shrinkage parameter $s$ from the set $M^{(i)}$. The remaining pixels in $M^{(i)}$ are entered into the next iteration. The algorithm converges until the BS index of the extracted image satisfies the desired level. Finally, we can get a smooth, conservative image boundary around the core lesion area.

## 3 EXPERIMENT: THE LYMPHATIC MALFORMATION LESIONS TUMOR BOUNDARY EXTRACTION

The Lymphatic malformation disease is a benign tumor, so it requires that the healthy tissues are well retained in the resection surgery. In order to make the boundary easy to execute by machine,

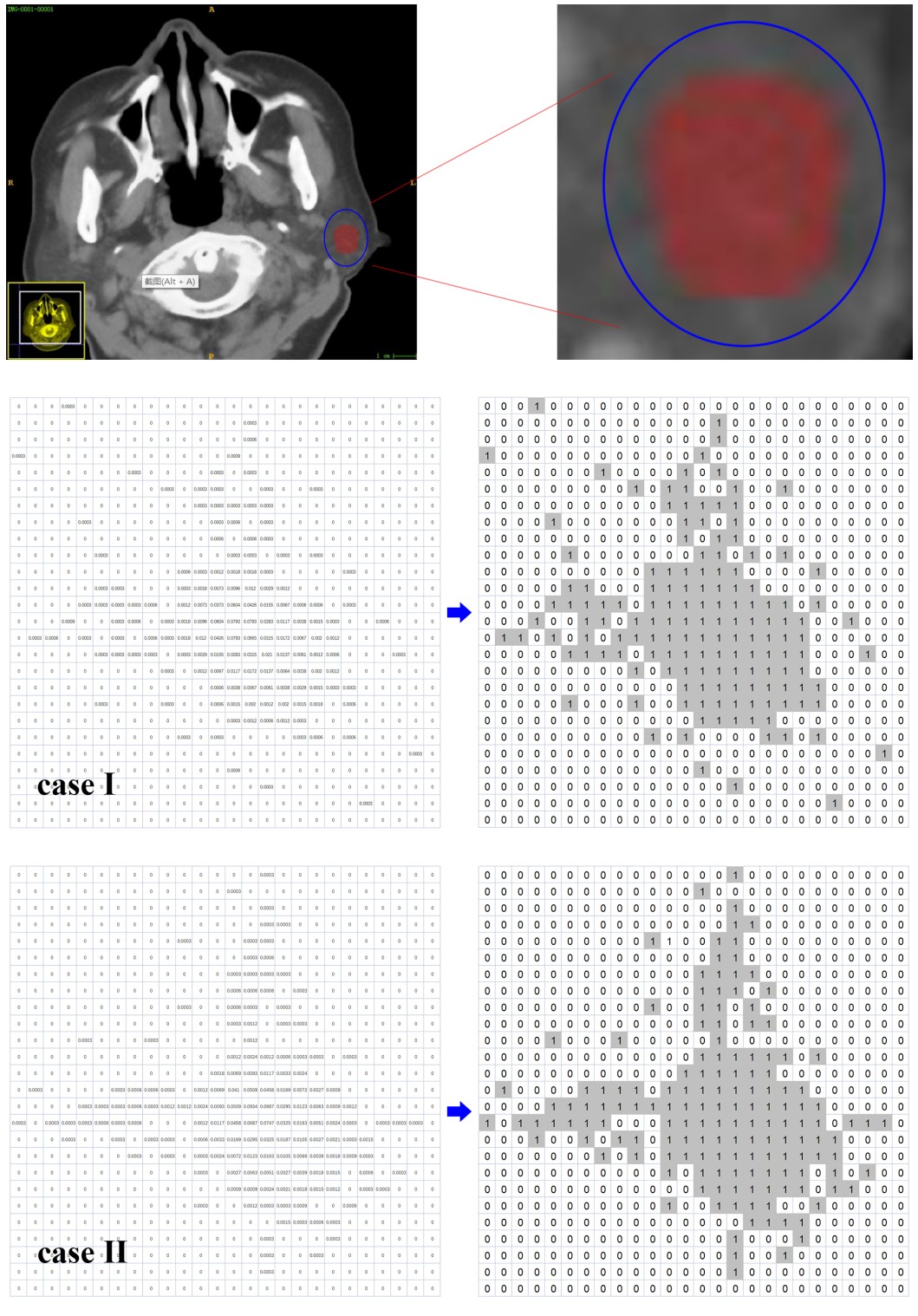

Figure 2: The CT images and data transformation of the ILM tumors. Top panels: a CT image example of the ILM tumor with the target area marked in red manually. Middle & bottom panels: the data transformation of two randomly selected CT image cases, where the left panels are the original grayscale CT images and the right panels are the transformed 0-1 matrices of the CT image.

---

**Algorithm 1:** CIBE Algorithm

---

**Input:** A three-dimensional CT image $M$, the hyperparameters $k$, $\lambda$, and $s$

**Output:** A CT image with conservative extracted boundary $\widehat{M}$

**Step1** Transform the three-dimensional CT image data into a 0-1 matrix by
$f : z|x, y \to \{0, 1\}$.

**Step2** Iteratively update $M^{(i)} = M_1^{(i)} \cup M_0^{(i)}$. At the $i^{th}$ iteration, $\forall (x, y) \in M_1^{(i)}$ **do**

    **Step2.1** Obtain the set of $k$-nearest neighbors $KNB(x, y)$,

$$KNB(x, y) = \{(x'_j, y'_j) \,|(x'_j, y'_j) \in M \text{ with the smallest}$$
$$d_{(x,y),(x',y')}, \; j = 1, \ldots, k\}.$$

    **Step2.2** Calculate the Pixel Deepness $PD(x, y)$,

$$PD(x, y) = \frac{\sum_{(x',y') \in KNB(x,y)} FD(x', y') \left( \frac{1}{d^2_{(x,y),(x',y')}} \right)}{\sum_{(x',y') \in KNB(x,y)} \left( \frac{1}{d^2_{(x,y),(x',y')}} \right)}.$$

    **Step2.3** Calculate the boundary smoothness of $M_1^{(i)}$,

$$BS(M_1^{(i)}) = \min_{(x,y) \in M_1^{(i)}} PD(x, y).$$

    **Step2.4** Verify the smoothness level of the boundary,

        **if** $BS(M_1^{(i)}) \le s$ **then**

            remove the redundant points from $M^{(i)}$ in the following way:
            $\forall (x_0, y_0) \in M_1^{(i)}$, if $PD(x_0, y_0) < s$, set $f(z|x_0, y_0) = 0$.
            Therefore,
$$M_1^{(i+1)} = M_1^{(i)} - (x_0, y_0),$$
$$M_0^{(i+1)} = M_0^{(i)} + (x_0, y_0);$$

        **else**
            return $\widehat{M} = M^{(i)}$.

---

the calculated boundary must be smooth enough. The data used in this article is a set of sliced CT diagrams of the lymphatic malformation lesions. For each patient, there are 13 CT images. As shown in the top panels of Figure 2, for each patient, physicians first identify the lesion area in the CT image and manually mark it in red based on their experiences. The left side panel is the original CT image with the target area marked in red manually by the surgeons. The right side panel is the amplificatory image of the red area of the image in the left panel. In our method, the CT image is converted to a 0-1 matrix at the first step. The middle and bottom panels of Figure 2 are two randomly selected CT image cases. The left side panels are the original CT images. The right side panels are the converted CT images.

Before applying the proposed CIBE algorithm, we need to determine the hyperparameters $k$, $\lambda$ and $s$, where $k$ is the number of the nearest neighbors, $\lambda$ is the sensitivity parameter of $FD$, and $s$ is the shrinkage parameter. We use the Case-I as an example to illustrate the process of tuning the hyperparameters. To facilitate the observation of the impact of different parameters, only one parameter is tuned at a time with the rest parameters to be their default levels. The default settings of these parameters are $k = 20$, $s = 0.45$, and $\lambda = 0.5$. We test these parameters at different levels, which are $k = \{4, 8, 12, 16, 20, 24, 28, 32, 36\}$, $\lambda = \{0, 0.125, 0.25, 0.375, 0.5, 0.625, 0.75, 0.875, 1\}$, and $s = \{0.1, 0.3, 0.4, 0.42, 0.44, 0.45, 0.46, 0.47, 0.48\}$. Due to the limitation of space, we only choose four tuning results for each parameter and show them in Figure 3. The left panels of Figure 3 are the extracted boundaries under different levels of $k$. The extracted boundaries are quite similar to each other for $k \le 28$. In our experiments, if $k \ge 32$, the core area will be cleared out completely.

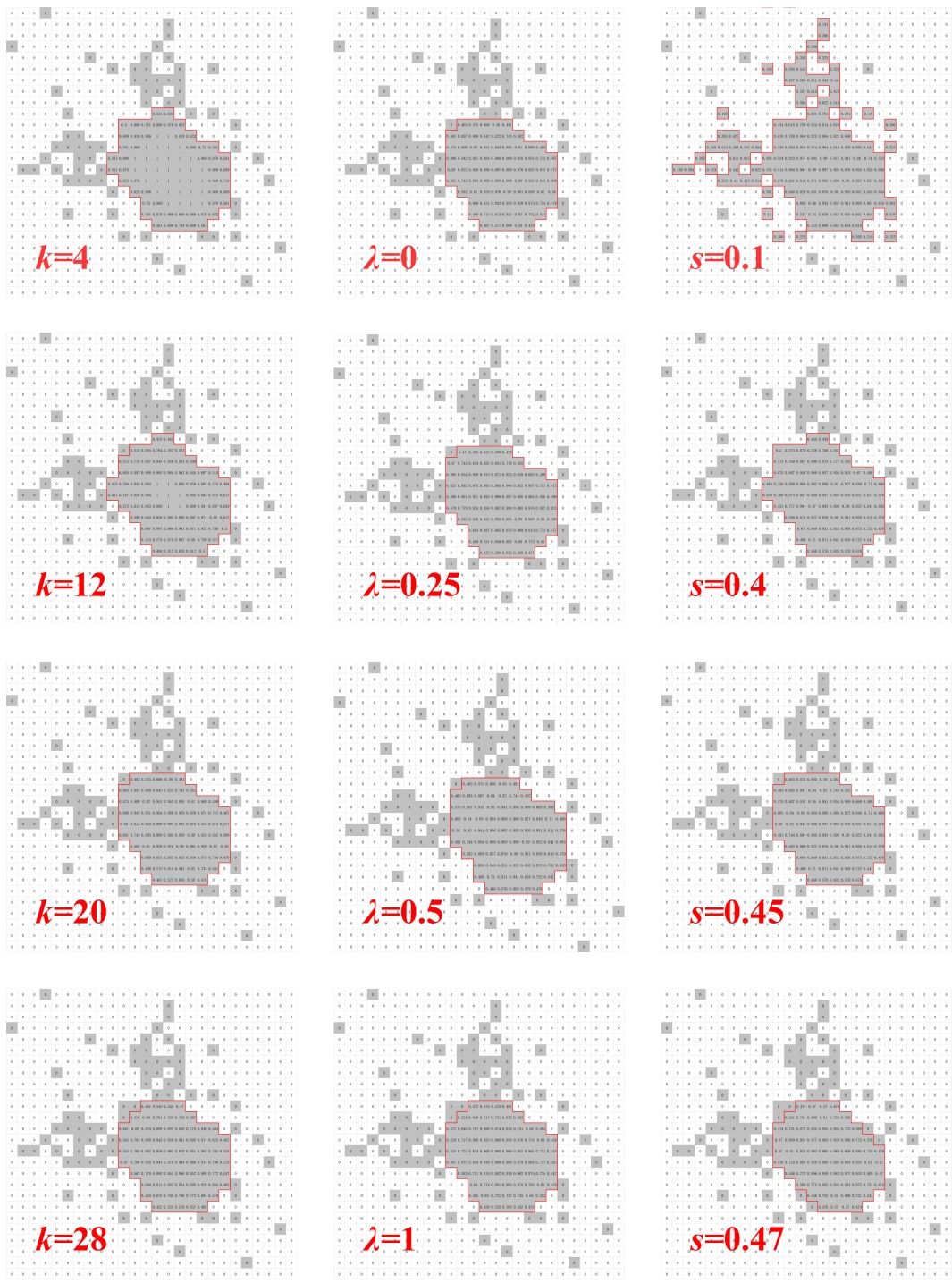

Figure 3: The tuning results of the hyperparameters for Case-I. Left panels: the results of boundary extraction under different values of the parameter $k$. Middle panels: the results of boundary extraction under different values of the parameter $\lambda$. Right panels: the results of boundary extraction under different values of the shrinkage parameter $s$.

173 Combing the results of all case studies together, we find $k = 20$ is a reasonable choice, and the
174 result is also quite stable. The middle panels of Figure 3 illustrate the extracted boundaries under
175 different values of $\lambda$. From the figure, we see that varying $\lambda$ from 0 to 1 has no significant impact
176 to the results, which indicates that in the ILM tumor boundary extraction experiments, $\lambda$ is not a
177 critical parameter. We choose $\lambda = 0.5$, that is the $FD$ index is affected by itself and its $k$ neighbors
178 equally likely. The right panels of Figure 3 are the boundary extraction results under different values
179 of the shrinkage parameter $s$. From the figure we can see that when $s \geq 0.48$, the image is cleared
180 out completely. In our experiments, for $s = 0.45$, the CIBE algorithm will provide a satisfactory
181 boundary extraction result.

## 3.1 BOUNDARY EXTRACTION RESULTS

183 From the results of the previous section, the appropriate hyperparameter setups are $k = 20$, $s = $
184 0.45, and $\lambda = 0.5$. Figure 4 presents the boundary extraction results of these two randomly selected
185 ILM tumor images in Figure 2 using the proposed CIBE algorithm.

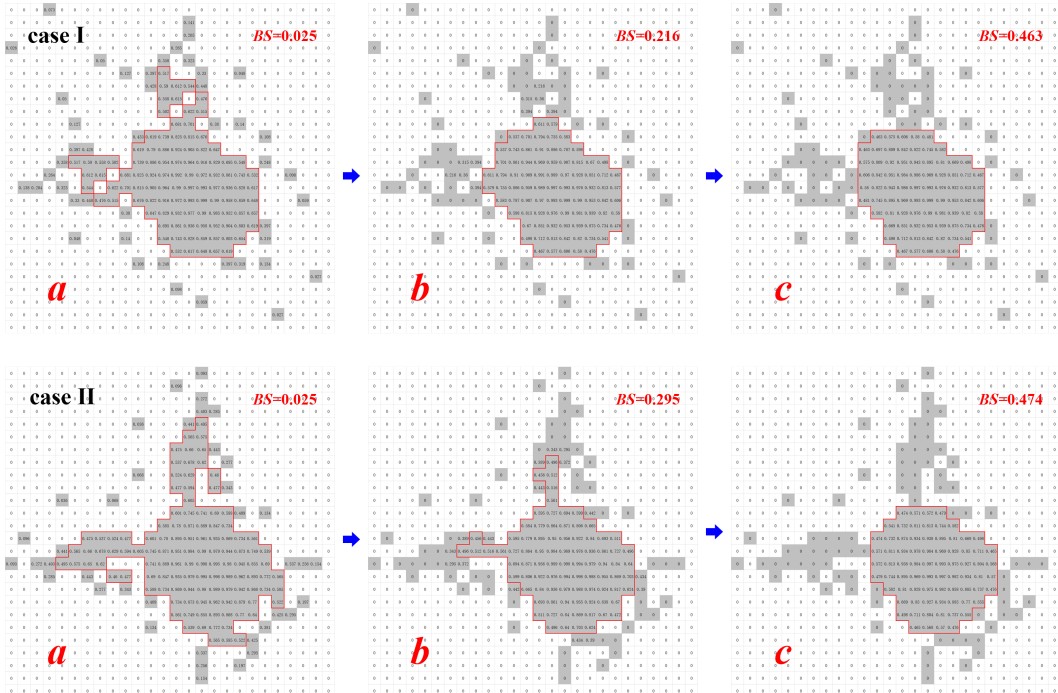

Figure 4: The boundary shrinkage process diagrams for these two cases mentioned in Figure 2. Top
panels: Case-I. Bottom panels: Case-II. For each case, panel (a) is the result of the CIBE algorithm
in the first round; panel (b) is the result of the CIBE algorithm in the third round; panel (c) is the
final extracted boundary of the CIBE algorithm. The gray points are the locations of the original
1-points; the red lines are the extracted boundaries; the gray points outside the red boundary are the
discarded noise points; the values on the pixels inside the red boundary indicate their PD indexes.

186 Let's use Case-I as an example to illustrate the whole boundary extraction process of our CIBE
187 algorithm, which is shown in the top panels of Figure 4. In the first iteration, the CIBE algorithm
188 calculates the $PD$ indexes of all class-1 points and then the BS index of the image. CIBE removes
189 all points with their $PD$ indexes lower than the shrinkage parameter $s = 0.45$ out of the class-1 area.
190 The extracted boundary and the estimated $PD$ indexes after the first iteration are shown in Panel
191 (a). The $BS$ index of the current boundary is 0.025, which is less than the shrinkage parameter
192 $s = 0.45$. The algorithm moves to the next iteration. Panel (b) is the boundary extraction result
193 after 3 iterations. At the final iteration, the $BS$ index of the current boundary is 0.463, which is

greater than the shrinkage parameter $s = 0.45$, the algorithm converges and return the final extracted conservative and smooth boundary image as shown in Panel (c). The final extracted boundaries (the red lines) of these two randomly selected cases are sufficiently smooth and conservative. The extracted boundaries are very accurate, which cover the core lesion areas properly, and do not contain any healthy tissue. Successful application of our CIBE algorithm in the resection operations allows to remove the core lesion areas of the ILM tumors without contacting any healthy tissues.

## 4 CONCLUSION

This article investigates the boundary extraction problem with special requirements. The whole target area is no longer required to be within the boundary in the ILM tumor resection surgery. The redundant points away from the core part of the target area have to be abandoned with the request that the boundary is as smooth as possible. However, the existing methods are not able to achieve this goal. Therefore, we proposed a conservative boundary extraction method with a well well-designed iterative boundary shrinkage procedure that can remove the redundant points around the lesion area and obtain a smooth and conservative boundary for the core part of the CT image. The results shows that the CIBE method provides a smooth boundary of the core lesion area of the ILM tumor properly. Although the proposed CIBE algorithm is developed in the two dimensional space, it can be easily extended to the three dimensional scenarios. As the development of the artificial intelligence, the proposed CIBE algorithm will be useful if integrated with the robotic surgeries which require a conservative boundary extraction in the lesion areas.

AUTHOR CONTRIBUTIONS

ACKNOWLEDGMENTS

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
