# OpenReview forum: "A Conservative Image Boundary Extraction Method with Application to the ILM Tumor Surgery"
_ICLR.cc/2024/Conference — ICLR 2024 Conference Withdrawn Submission_

### Official Review · Reviewer_bXZY · 2023-10-30

**Soundness:** 1 poor
**Presentation:** 2 fair
**Contribution:** 1 poor
**Rating:** 3
**Confidence:** 4

**Summary:**

The author proposed a rule-based image segmentation algorithm for 2D CT images with lymphatic malformation lesions. Three measurements are designed for distinguishing the lesion regions from the background, i.e., fuzzy degree, pixel deepness, and boundary smoothness. The lesion region is segmented by performing an iterative region shrinkage-based approach with repeated measurements of regions. 12 CT images are adopted for the experiments, and no comparison results are reported with other methods. The presented work is overall preliminary and not ready for publication.

**Strengths:**

+ Looking into detailed features of a particular disease
+ The description of some typical cases with detailed illustrations of region shrinkage is provided.

**Weaknesses:**

- The presented approach is designed largely based on the specific knowledge of the target lesion and is hard to generalize to other segmentation targets. How will the proposed method work on non-round targets?
- The motivation for introducing the three measurements is not clearly introduced. What's the meaning of these three measurements?
- Is the method 2D or 3D? CTs are 3D;  how the extra dimension is handled, and what will be the benefit of considering them in 3D?
- The experimental dataset is too small, i.e., 12 CT images. It is hard to appreciate the advantages of adopting the proposed method.
- Previous segmentation methods are not compared in the experiments.
- The presentation of the work is not sound, e.g., figures are blurring(Figure 1), equations are not numbered, functions are used before defined (f in Definition 1), quantitative results are missing, etc.

**Questions:**

None

---

### Official Review · Reviewer_v2GP · 2023-11-01

**Soundness:** 1 poor
**Presentation:** 2 fair
**Contribution:** 1 poor
**Rating:** 1
**Confidence:** 4

**Summary:**

This work presents a method for identifying a suitable conservative boundary based on annotated CT image slices of infant lymphatic malformation. The purpose of the conservative boundary is to aid surgical decision making.

The method is an iterative scheme applied to a boundary extracted from a binary image to refine a suitable area for surgical removal.

**Strengths:**

The problem itself seems of interest although outside the scope of this reviewer’s expertise.

The methods developed seem appropriate and are relatively well described.

**Weaknesses:**

Little is said about the dataset itself and the method in which the annotated greyscale CT image slide is turned into a binary mask. Basic information like the number of patients appears to be missing.

There is very little prior art referenced in the work which suggests that more could have been done to compare against either baseline methods or, alternatively, approaches used in other contouring settings such as those used for radiotherapy planning.

This work has promise, however may be better suited for a medical image analysis venue rather than ICLR – it has little to do with representation learning.

**Questions:**

Line 42: More detailed explanation of the annotation process is required. Why does the annotation render the raw pixel values meaningless? A diagram to explain may help.

Line 72: Is this not a 2D image? May be better described as a 2D function of x, y -> z. I.e. any point in x, y maps onto the pixel value z.

**Details Of Ethics Concerns:**

No information is given on where the data has been collected from an under what basis. Whilst they are non-identifiable images, there may be ethics or data governance requirements depending on the institution and territory in which the work was undertaken.

---

### Official Review · Reviewer_4CWF · 2023-11-04

**Soundness:** 2 fair
**Presentation:** 2 fair
**Contribution:** 1 poor
**Rating:** 3
**Confidence:** 4

**Summary:**

This paper is about ILM tumor boundary detection in CT images and the extracted boundary must be conservative covering the core part of the lesion area. The new approach, namely the conservative image boundary extraction (CIBE) approach, uses the concepts of Fuzzy Degree, Pixel Deepness and Boundary Smoothness, and the iterative boundary shrinkage procedures.

**Strengths:**

New concepts and measures are defined for boundary detection.

**Weaknesses:**

Although the new measures (FD, PD and BS) are defined, the 2D experimental results are mostly qualitative. More experiments with quantitative analysis are expected to show the properties of the proposed method. No comparison with other related, state-of-the-art methods is shown.

**Questions:**

1.	The boundary is always shrinking to find the ILM tumor. Any particular reason for not including expansion in the framework?

2.	Fuzzy degree describes the exposure level of a point in class M. The exposure level needs more descriptions.

**Details Of Ethics Concerns:**

N.A.